# OpenReview forum: "Holistic Evaluation of Language Models"
_TMLR — Accepted by TMLR_

### Review · Reviewer_JcuA · 2023-04-01

**Summary Of Contributions:**

INTRODUCTION

This paper presents a "holistic evaluation of language models". It defines "scenarios" (task-dataset configurations) and metrics, which comprise an abstract taxonomy for evaluation.  Both scenarios and metrics are generally described in terms of their properties: for example, scenarios are defined by a task, "what/who/when", language, etc.  Considered in this light, existing benchmarks fall well short of covering the entire cross-product of possible configurations, or even having *any* task to represent many groups of users.

For each scenario that is chosen, the paper tries to be as thorough as possible across seven evaluation axes: Accuracy, Calibration, Robustness, Fairness, Bias, Toxicity, and Efficiency.  Notably, this comprehensive evaluation significantly extends the coverage of aspects like calibration and fairness compared to prior focused evaluations.

This paper considers 30 LMs; some of these have never been publicly evaluated, and many on only a small subset of the scenarios considered here.  Specifically, the paper claims to go from 17.9% to 96% coverage of scenarios, with 87.5% coverage of (scenario, metric) pairs. Ultimately there are 30 LM evaluations on 42 scenarios (16 core and 26 additional targeted ones).

25 core findings are listed. In my opinion, the most surprising and impactful ones that differ or extend from prior work (at least as I'm aware) are:

4. relationship between robustness/fairness/accuracy

5. performance disparities across features like dialect

12. toxicity detection

14. linguistic understanding and how it differs from other tasks

17. memorization

19. BBQ (other analysis of BBQ I've seen largely post-dates this paper)

21. while "obvious", some of these findings, including the highlighted NarrativeQA one, are interesting

24. perplexity; this is particularly valuable in the context of a large-scale study compared to past scaling laws work

Points like 16 have been evaluated very extensively (e.g., in the context of self-consistency https://arxiv.org/pdf/2203.11171.pdf and many follow-on papers for chain-of-thought for mathematical reasoning). However, I grant that the time taken to prepare work like this makes it hard for all conclusions to remain fresh.

CONTRIBUTIONS

1. Taxonomy, leading to broad coverage and recognition of incompleteness

2. Multi-metric measurement and evaluation of many existing models

3. Results

4. Interactive results and codebase

SUMMARY OF THE PAPER

I will now go through the paper section-by-section and interleave some comments here while mostly reserving judgment for the strengths + weaknesses section.

Sections 2 and 3 present general definitions for the framework and define the scenarios, which borrow both from tasks corresponding to tracks at ACL conferences and also from rising applications of LLMs for more "unusual" tasks. It also sets up the restriction to dialects of English rather than other languages, as well as the set of tasks that are to be used.

The later sections of 3 (3.3 onwards) then describe the selection of benchmarks from standard datasets used in the evaluation here. Of note is that IR is framed as a task of scoring individual candidate passages using an LM, a relatively less-studied setting. The tasks are discussed with appropriate caveats (e.g., disconnect of toxicity from real settings, the inability to cover every conceivable QA setting, then issues with CNN/DM and XSum for summarization).  Nevertheless, the paper strikes a balance between its ideals and its pragmatic constraints, selecting reasonably popular datasets that are representative of the different scenarios.

Section 4 then describes metrics. As with tasks, these are compiled down from "well-studied" metrics across communities spanning NLP, ML, AI, fairness, data mining, and more.  Only six of these criteria are used in the current study: accuracy, bias, fairness, inference efficiency, robustness, toxicity, and uncertainty/calibration. This set is still quite a bit broader than those typically considered; for example, calibration could be evaluated in a large fraction of NLP papers, but usually isn't.  The paper rationalizes which set of metrics is used for which scenarios. Of note, selective classification is also explored for calibration (in addition to the typical ECE). Robustness is explored in the contexts of invariance (using NL-Augmenter) and equivariance (contrast sets), with other forms of distribution shift like subpopulation shift omitted due to impracticality. Fairness is similarly evaluated with counterfactual fairness (perturbation-based) as well as performance disparities, which require group-level metadata. Bias and toxicity are measured in standard ways. Finally, efficiency is measured in terms of energy and CO2, both for training and inference, using either previously reported values or new estimates.

Section 5 presents evaluations. 5.1 presents language modeling settings, both on standard and non-standard datasets (e.g., TwitterAAE) as well as minimal pair evaluation.  The discussion of dialect surrounding TwitterAAE and ICE here seems carefully done and insightful (but I am not an expert in this area).  5.2 presents tests around knowledge, including QA (largely on standard datasets) and a fact-completion task like LAMA, but curated to be more broad.

5.3 presents reasoning tasks. These are less standard. The division of tasks is interesting and grounds in prior literature.  Four "primitive" settings (including synthetic data like Dyck languages and bAbI) and five realistic settings are presented.  This includes a novel LegalSupport task. 5.4 discusses copyright, and tests whether models can produce completions of books or the Linux kernel. 5.5 explores narrative reiteration (generating headlines) and narrative wedging (dividing groups based on group identity). These follow on datasets and settings introduced in past work (Gabriel et al. and Buchanan et al.). Finally, bias is measured on BBQ and toxicity in BOLD and RealToxicityPrompts.

Section 6 describes models, including size and cost. This is a useful aggregate resource in terms of seeing tokenizers and parameter counts all in one place.  There's discussion of versioning and risks of things like models changing during the evaluation itself or the versions they report becoming deprecated.  The discussion of contamination is also useful as the conversation around that continues to evolve with newer models.

Section 7 describes prompting, including an abstract format presented in Table 7. Appendix I does a good job of situating this prompting with respect to prior prompt repositories.

Section 8 describes results.  8.1 focuses on inter-metric relationships, such as accuracy vs. robustness or accuracy vs. calibration. Accuracy, robustness, and fairness are all quite correlated, which is particularly surprising in the case of fairness. Correlations are also discussed on each scenario. Figure 26 reports the main "bake-off" results across all models and metrics in a head-to-head comparison. No model is a clear winner; while text-davinci-002 is ahead on accuracy, its calibration is poor, for example. Still, some broad trends about bigger models and "better trained" (more recent models often from bigger labs) are commented on.

Openness is also discussed, with a comparison between open/limited/closed access models. I do feel that this comparison is a bit skewed based on the small number of models that happen to exist (e.g., OpenAI is an outlier along many dimensions, so their choices impact this a lot) and may not be indicative of broad trends in these areas. Finally, size and perplexity and their relationship with accuracy across models are discussed.

8.2 looks at variance across prompts and number of in-context examples. The number of examples is quite interesting in that many tasks exhibit a big jump from zero-shot to one-shot, but little improvement after that, contradicting the conventional wisdom about more examples being better. Prompt formatting experiments are not visible in this version of the paper.  Variance in multiple-choice prompts is also reported, again with variance, where the best prompting approach varies across tasks and across models (with Anthropic being different from the rest of the pack).

8.3 shares results separated by task. Some trends are commented on but the only really surprising result is the toxicity on NarrativeQA, which the paper says may be due to false positives from the Perspective API. Scaling issues and underperformance of IR are discussed. RAFT shows interesting and varied performance across its subtasks, with text-davinci-002 doing 50% absolute worse than GLM on the Systematic Review Inclusion subtask.

8.4 looks at BPB on different datasets for targeted evaluation of linguistic competence. TwitterAAE shows that all models do poorly on AAE compared to White English.  Knowledge doesn't reveal much, but there is large variance on reasoning, with many systems getting very low accuracy on math settings. bAbI / Dyck show a nice spread across different models. Some of the largest models can regurgitate copyrighted text, but only infrequently. For toxicity, most models are not toxic unless a toxic context is given. The most surprising finding in this section is that most models demonstrate the opposite of expected biases on BBQ in ambiguous contexts.

8.5 conducts human evaluation. In disinformation, reiteration works well under most models but wedging is a challenge, with mixed results from most models.

The related work provides a sweeping and (in my view) unnecessarily grandiose overview of both modeling and benchmarking.

Section 10 describes a form of limitations: "what's missing". The discussion of missing scenarios highlights the strengths and weaknesses of HELM. Evaluations, models, and adaptation methods are all missing for somewhat clear reasons (not everything can be covered in a fast-moving field). Finally, Section 11 describes other limitations and contextualizes the work.

**Audience:**

Yes

**Broader Impact Concerns:**

There is not explicitly a Broader Impact statement, but the paper devotes much of its real estate to broader impacts in both explicit and implicit ways.  The largest broader impact of this work will be from a research sociology point of view in how it shapes the field, and I feel ill-equipped to comment on that.  Certainly the discussion in the paper is thorough enough and carefully contextualizes the claims.

**Claims And Evidence:**

Yes

**Requested Changes:**

I will make some recommendations here, but I am not proposing large changes required for acceptance. The cost of running the experiments is such that it seems impractical to redo anything significant.  Moreover, I think there is little value in suggesting a lot of work since much of the community has read this paper already. Its numbers may become gradually more and more out of date, but the right patch there is follow-on work or a website or something, not silently updating the PDF with ChatGPT and replacing every win of davinci-002 with a win of davinci-003.

Medium changes:

Figure 2/8 are slightly confusing to me in their current form. The first column, tasks, are basically mutually exclusive. But the third column is not; these are features of the population (e.g., you could have Black women web users as a condition for a dataset). The "lines" in this diagram therefore don't seem like a complete way of specifying a scenario.

Minor changes:

p5. prompt decomposition should be followed by citep not citet. Same with Welbl on p20 and Persily on p21.

Section 5.1: "To measure a model's understanding of the English language" -- "understanding" is a very loaded term. I recognize that the rest of this paragraph clarifies exactly what is meant here.  But I still might soften or rephrase this.

p42. "AI Labs" -> "AI21 Labs"

p49. I think this graph is a bit confusing. The caption should make it clear that this is the max accuracy (SOTA) over all previous models up to that timestep. Otherwise it's easy to look at GLM and think that it's the best.

p50. work has also show -> work has also shown

p51. Figure 29 does not have subplots? This kind of makes that section hard to interpret as we only have the cumulative plot here.

**Strengths And Weaknesses:**

STRENGTHS

In terms of scope, this really is a first-of-its-kind effort. It clearly advances the state-of-the-art of benchmark suites in terms of the careful consideration given to benchmark design and in terms of scale. The vast majority of language modeling work from OpenAI and Google only compares to papers within those two organizations and perhaps some printed numbers from other sources. I have seen no benchmarking of Cohere, AI21, and Yandex in any past work (which this paper also notes on page 5), let alone a comparison of all of these. And other models like GLM or UL2 are evaluated in niche subcommunities (or by the authors of those papers), but not as frequently in "premiere" language modeling settings. So this paper clearly fills a niche in evaluation that has not existed previously.

The motivation for the selection of scenarios, tasks, and domains, is generally good. Much of it is guided by pragmatic choices, and as a result it's a bit closer to reality than the ideal that the paper presents. But I also don't think it's very easy to see how this can be improved without additional dataset collection efforts or simply doing more (e.g., including less user-facing tasks like semantic parsing). The inclusion of RAFT and toxicity are particularly nice, as these are representative of a "new wave" of applications even though they don't have as long a history as many other tasks and datasets.

Reiterating one of the points above, I think the main broad benefit of this paper may be the clarity of argumentation around the benchmarking choices. Again, it is a major step up from GLUE and even efforts like BIG-Bench, which are more bottom-up and have some rough edges as a result. By showing a high attention to detail, I think this paper sets a standard for what is possible that hopefully others will emulate. Already, for example, I think we can see that the GPT-4 paper's evaluation looks narrow and crappy according to many dimensions of this paper, whereas I'm not sure that narrowness would be quite as stark if this paer did not exist.

The interactive browser of the results is useful. I think its utility will be seen in how future work manages to use it; to me, many of the tables (both in the paper and there) strike me as "yep, that's a bunch of numbers", but I would guess that for almost any results reported here, practitioners in a specific area might find it very useful.

The acknowledgments and streamlining of enabling future researchers to cite the datasets and evaluations in HELM sets a nice trend that future work will hopefully continue!

Overall, I think this paper makes very clear what its contributions are and what the limitations are. In that sense, it is a very good fit for TMLR. The claims it makes are convincingly supported.

WEAKNESSES

Most of my weaknesses are more philosophical and I'm not sure they're easy to address. I share them in case they provide a useful perspective on the work.

The definition of "scenario" is a bit strange. In particular, I think the notion of "references" is fine, but Figure 6 really specializes these to the multiple choice domain, which I think is a mistake.  I think over-emphasis on multiple-choice is a major limitation of current work in the field.

"Adaptation strategies" seems increasingly important given the rise of RLHF models, and this is probably where this paper is the weakest. I recognize it is not practical to separately engineer every model, so I do not have a very concrete suggestion here.  But this is perhaps a weakness of an "academic" meta-evaluation as opposed to a stress-test in real settings. The strength of ChatGPT is that there is a whole ecosystem of users who *do* find out how to elicit the most powerful behavior, and findings like "let's think step by step" that are fit to notable models contribute to a kind of "attention lottery" (c.f. hardware lottery) where the results may bias towards more notable models like davinci-002. This seems to be underscored by the OPT divergences in multiple-choice prompting.

The experiments are very extensive to the point where they become hard to navigate. I don't think the paper should be penalized for this. It's hard as a single reviewer to say what results should be prioritized.  But because of the length of this paper and the sheer number of results, I feel that some very interesting gems are easy to overlook. For example, I think Figure 32 is a very interesting result and quite relevant to lots of NLP practitioners: knowing how many examples are optimal / needed is pretty important for a range of applications. But more space is devoted to Figure 33 on multiple-choice questions and their discussion. I'm acknowledging my own personal bias here, but multiple-choice questions are maybe the worst offender in terms of a task that LLM designers care about but which have little practical value for real applications. How to overfit the format is supremely uninteresting in my view.  I think the question of how to format results like this to make them maximally accessible is an interesting one, and reading the paper in a linear fashion as a reviewer is certainly not the right way to consume it. The presence of the interactive visualization alleviates this weakness to a large extent.

---

> ### Author Response · Authors · 2023-04-27
> **Response to Review JcuA**
>
> Thanks for the thorough and precise review, we appreciate it, especially given the length of this paper! We address each of the weaknesses and requested changes below.
>
> *Weaknesses.*
> > The definition of "scenario" is a bit strange. In particular, I think the notion of "references" is fine, but Figure 6 really specializes these to the multiple choice domain, which I think is a mistake. I think over-emphasis on multiple-choice is a major limitation of current work in the field.
>
> We agree the field often fixates on multiple-choice, which we see as a convenience for evaluation design/execution but quite restrictive/lacking in ecological validity in most contexts. With that said, Figure 6 is merely intended to illustrative for the general audience. For example, tasks like summarization (where softer matching against reference is needed) or story generation (no reference) are also fine. Happy to update the wording if this feels unclear as presented.
>
> > "Adaptation strategies" seems increasingly important given the rise of RLHF models, and this is probably where this paper is the weakest. ...
>
> We take this as an excellent point, which we have tried to foreground repeatedly. We note much of work conducted in 2021 and early/mid 2022, so the importance of the prompting details were not yet well-known (much of work pre-dates CoT; the paper is released prior to ChatGPT + GPT-4). We hope future work, including future versions of HELM, can address this but for now, we recognize and foreground the weakness, but cannot address it unfortunately. We do highlight some ablations in 8.2.
>
> > The experiments are very extensive to the point where they become hard to navigate. I don't think the paper should be penalized for this. ...
>
> We agree, this is an unfortunate concern that we also realized as writing the work and preparing it for submission. For this reason, we really encourage folks to use interactive visualization + we provide a table of contents in public version (and will again, assuming TMLR EiCs are ok with it). While beyond of the scope of the paper, we also note that we actively working to improve the navigation and data visualization on the public interface.
>
> *Requested Changes.*
> > Figure 2/8 are slightly confusing to me in their current form. The first column, tasks, are basically mutually exclusive. But the third column is not; these are features of the population (e.g., you could have Black women web users as a condition for a dataset). The "lines" in this diagram therefore don't seem like a complete way of specifying a scenario.
>
> Good point - we agree the underlying categories for who (and also potentially what) do not cleanly partition the space (e.g. who is itself a combinatorial space over different factors like demographic variables in race, gender, age).
>
> For simplicity, we thought it was better to just have a single category for "who" since in many cases, we do not have full metadata to identify the intersectional information (e.g. we might know the data comes from predominantly Black speakers, but do not know breakdowns for race, age, etc.). Happy to add a clarification to the caption if needed, though we currently prefer the *over*simplification since the figure appears early on and therefore is aimed at being accessible to a broader non-technical audience.
>
> > Minor changes.
>
> We have addressed all of the minor changes, thank you!

---

### Review · Reviewer_4CcX · 2023-04-13

**Summary Of Contributions:**

This paper addresses the challenge of providing transparent, standardized and holistic evaluation of (large) language models and has made the following contributions.
1. The paper presents HELM, an evaluation suite built on top of public NLP datasets with a taxonomy of user-facing tasks and evaluation metrics. The motivation for selecting each dataset/metric is well explained.
2. The paper outlines the principles of holistic evaluation and uses them to guide the evaluation methodology. It also clearly articulates the limitation of the current evaluation design and experiments.
3. Using the proposed evaluation suite, the paper benchmarks 30 language modes originated from a wide range of training corpora, model architectures, model sizes and organizations. Through the "dense multi-metric measurement", the paper is the first to provide an integral evaluation of models for specific scenarios/tasks, while traditional NLP benchmarks typically focus on one or two types of metrics (e.g. accuracy or robustness). The results revealed to a great extent the relative strengths of the evaluated models (e.g. through the head-to-head win rate comparison across a number of tasks). The paper also conducted a series of thought-provoking analysis experiments (e.g. examining the correlation between accuracy and robustness metrics).


**Audience:**

Yes

**Broader Impact Concerns:**

The paper has two well-written sections ("What's missing" and "Limitations") and I don't have more concerns regarding its broader impact.

**Claims And Evidence:**

Yes

**Requested Changes:**

1. Please add a paper outline at the beginning.
2. Section 4.6 & 4.7 are very difficult to read. Especially I find it hard to understand what distinguishes "fairness" from "bias & stereotypes" and why they cannot all be merged as sub-metrics of the "fairness" category. Addressing this is critical to secure my recommendation.
3. Very likely you can shorten the presentation of section 3 & 5 by providing succinct summaries of the task/eval specifications plus 1-2 sentences of motivation (similar to a checklist), and move the full motivational discussion (why a particular paper is chosen) to the appendix in a format that people can easily index.
4. Can you include a few scenarios that are more close to real-world usage of language models (e.g. some interactive tasks that involves human testers)? This would address weaknesses 1 & 2 mentioned above and strengthen the work in my view.
5. Can you include more tasks with freshly created data that are guaranteed not seen by any of the LMs? This would address weakness 3 mentioned above and strengthen the work in my view.
6. Can you elaborate more on what engineering improvement you might introduce to the benchmark to make future model evaluation efficient and easy? This would address weakness 4 mentioned above and strengthen the work in my view.

**Strengths And Weaknesses:**

Strengths:
- The paper maximized the utility of public NLP datasets and the proposed evaluation methodology offers a fair amount of insights into the strengths and limitations of existing language models.
- Through "dense multi-metric measurement", the paper extend "beyond-accuracy" NLP evaluation to a much larger number of tasks comparing to existing work, which is crucial for LLM evaluation given scaling has updated the SOTA accuracy for a large number of existing benchmarks.
- Several analysis results from the paper are thought-provoking and may inspire future research, e.g. the relationship between model fairness/robustness and their accuracy on a particular task.
- This paper makes a great example of approaching LM technology from a rigorous scientific perspective instead of hype-chasing in a rapidly developing field.

Weaknesses:
- Unfortunately due to the rapid-growing nature of the field, the analysis in the paper feels somewhat disconnected with the state-of-the-art due to its heavy reliance on traditional NLP datasets. For example, it is hard for me to leverage the insights from any of the discussed scenarios to understand why GPT-4 can correctly respond to many of my everyday requests that involve sophisticated intent and long-context understanding (which is amazing and hard to imagine just two years ago). It is also difficult for me to predict if a model can potentially show any "emergent capability" (e.g. making birthday party plans) given the evaluation on the chosen tasks.
- While the paper added extensive motivation for task/metrics selection in each evaluation scenario, reading section 3 (Core Scenarios) and 5 (Targeted Evaluations) makes me feel that they are mechanical combinations of known NLP datasets. Besides, I often cannot tell whether there is a good reason to favor one dataset over the other in a particular scenario esp. when I'm not the expert in that domain (e.g. why choose IMDB as the representative dataset for sentiment analysis). This said, I think taxonomizing NLP tasks is intrinsically hard and the paper did a reasonable first step in choosing those tasks/metrics.
- Given some of the evaluated models are restricted/closed, the data contamination status is unclear for these models. This may impact the validity of the analysis (esp. the meta-analysis) in the paper, and I personally feel that I have to take many of the conclusions with a grain of salt given this.
- Another concern of mine is the actual execution and maintenance of the benchmark, given it is pretty costly to evaluate all models on all scenarios and all metrics.
- Finally, the paper is very long (which I wonder if it is necessary) and it is difficult for readers to digest the results and get useful insights.

---

> ### Author Response · Authors · 2023-04-27
> **Response to Review 4CcX (1/2)**
>
> Thanks for the thorough and precise review, we appreciate it, especially given the length of this paper! We address each of the weaknesses and requested changes below.
>
> *Weaknesses.*
> > Unfortunately due to the rapid-growing nature of the field, the analysis in the paper feels somewhat disconnected with the state-of-the-art due to its heavy reliance on traditional NLP datasets. ...
>
> This is an excellent point: we agree with the sentiment as this poses an important challenge for our field. We highlight two points
> 1. Much of this work was conducted in late 2021 and early 2022. The relevance of the benchmark selection arguably has not endured to the fullest extent design, but highlight the work was released prior to ChatGPT and GPT-4 (among other works) that cause us to change our evaluation priorities. Namely, how LMs would be used and the extent to which this diverges from standard NLP tasks was not yet clear at the time.
>
> 2. In the coming releases of HELM, we will add more such evaluations. While it is always tricky to keep pace and this is not a reflection of HELM the paper but the living benchmark, we do highlight this disconnect can and will be addressed.
>
> > While the paper added extensive motivation for task/metrics selection in each evaluation scenario, ...  feel that they are mechanical combinations of known NLP datasets. ...  cannot tell whether there is a good reason to favor one dataset over the other in a particular scenario
>
> We agree that it is hard to know how to purely justify taxonomies but, on balance, we feel the work takes a reasonable attempt and clearly advances the taxonomization known in the field, while being forthcoming in its own limitations.
>
> > Given some of the evaluated models are restricted/closed, the data contamination status is unclear
>
> We agree, but we have little recourse to address this at present. Independently, future work may develop new techniques for contamination analysis that could be applied retrospectively to HELM, but this is out of scope for this work itself.
>
> > Another concern of mine is the actual execution and maintenance of the benchmark, given it is pretty costly to evaluate all models on all scenarios and all metrics
>
> While fair, it is unclear how this should be better addressed given the range of models, desiderata, and scenarios of interest. We suspect future work could pare down the evaluation, e.g. find the "principal components" of HELM itself, i.e. casting evaluation as a constrained optimization problem of minimizing cost while maximizing the ability to generate results predictive of the full gold-standard HELM evaluation. We have outlined these possibilities in Appendix G by introducing a priority system with illustrative analogies to the knapsack problem and core set selections.
>
> *Requested Changes.*
> > Please add a paper outline at the beginning.
>
> We will add a table-of-contents assuming the EiCs for TMLR are ok with that.
>
> > Section 4.6 & 4.7 are very difficult to read. Especially I find it hard to understand what distinguishes "fairness" from "bias & stereotypes" ...
>
> We recognize the terms are often used exchangeable, though we believe that there is a clear distinction we explain below. For terminology, we follow the conventions of literatures usually referenced to as "fairness in ML/algorithmic fairness" and "bias in NLP" (e.g. see the names of the workshops usually collocated at NeurIPS/ICML like FAT* vs. those collocated at ACL/EMNLP like Gender Bias in Natural Language Processing).
>
> *The critical distinction* is the questions of fairness refer to disparities *in the presence of desired outcomes*, i.e. the performance of the model when compared to the ground truth answer. The specific operationalizations align with the well-established notions of individual and group fairness.
>
> In contrast, the questions of bias refer to disparities *in the absence of desired outcomes*, i.e. the performance of the model without any consideration for the ground truth answer. The specific operationalizations align with well-established notions of erasure/overrepresentation and stereotypical associations.
>
> Ultimately, this means our measures of fairness are best operationalized for classification (or classification-esque) scenarios, where we can ask if the model is similarly correct across groups (i.e. fairness is computing accuracy on a specific subset). Whereas the measures of bias are best operationalized for generative scenarios, where we can if the model is similarly likely to mention different groups (i.e. bias is not computing any notion of accuracy).

---

> > ### Author Response · Authors · 2023-04-27
> > **Response to Review 4CcX (2/2)**
> >
> > *Requested Changes (continued).*
> > > Very likely you can shorten the presentation of section 3 & 5 by providing succinct summaries of the task/eval specifications plus 1-2 sentences of motivation, and move the full motivational discussion (why a particular paper is chosen) to the appendix in a format that people can easily index
> >
> > We acknowledge this is a matter of taste, but ultimately currently are still inclined to keep the extended discussion when each scenario is introduced. we prefer to keep all of the motivation alongside the dataset, because *we do want to drive the precedent that benchmarks should overtly reason about the many decisions involved* to ensure the operationalization tracks the conceptualization.
> >
> > We also highlight that the paper will contain both (i) a table-of-contents as requested and (ii) URLs that re-direct to individual (sub)sections of the PDF, which should ensure navigation/reference can be done successfully.
> >
> > > Can you include a few scenarios that are more close to real-world usage of language models (e.g. some interactive tasks that involves human testers)? This would address weaknesses 1 & 2 mentioned above and strengthen the work in my view.
> >
> > Adding new evaluations is not in scope at this time given the resources (engineering, money, model access) required. We will add these evaluations in the future to HELM (and have already since its initial release), but we see this as more relevant to HELM the living artifact than HELM the paper.
> >
> > We do highlight related work that builds on the HELM infrastructure to conduct precisely these new evaluations using extensive human evaluations: [Evaluating Human-Language Model Interaction](https://arxiv.org/abs/2212.09746)
> >
> > > Can you elaborate more on what engineering improvement you might introduce to the benchmark to make future model evaluation efficient and easy? This would address weakness 4 mentioned above and strengthen the work in my view
> >
> > We want to openly acknowledge that maintenance, etc. is beyond the scope of the paper, but we want to highlight our ongoing efforts, because it is difficult to disentangle the paper from the living artifact.
> >
> > The HELM benchmark is actively maintained by an engineering team at present. There is extensive documentation and we are collaborating with a range of organizations to ensure HELM is a widely-adopted standard.
> >
> > We refrain from saying more here because it is unclear how to do so without violating anonymity, but we are happy to clarify any details. In short, HELM's accessibility and efficiency has significantly improved and remains an area of active development, including through open-source and partner-based collaboration on its codebase.

---

> > > ### Comment · Reviewer_4CcX · 2023-07-05
> > > **Thank you for the thoughtful responses**
> > >
> > > The responses well addressed my concerns for the paper. I find it reasonable to leave adding new scenarios out of the scope of this work and I appreciate the highlighting of ongoing engineering efforts to maintain this benchmark. I strongly encourage you to add the limitations discussed in their rebuttal to the paper camera-ready wherever applicable.

---

### Review · Reviewer_gN7d · 2023-04-16

**Summary Of Contributions:**

The authors present an innovative approach, Holistic Evaluation of Language Models (HELM), which aims to enhance the transparency and understanding of large language models through comprehensive evaluation across various scenarios and metrics. HELM's abstract taxonomy allows for core and targeted assessments of 30 widely-used language models, covering accuracy, calibration, robustness, fairness, bias, toxicity, and efficiency in 16 core scenarios. By evaluating models under standardized conditions, HELM facilitates direct comparisons among them.

Some key findings from the HELM evaluation are:
- Instruction-tuning proves advantageous for large-scale language models by exploiting their reasoning or learning capabilities.
- A marked gap in performance persists between open-domain and closed-domain models, underscoring differences in task-specific formalisms and practices.
- The relationship between accuracy and calibration is nuanced as it depends on the assessment scenario and adaptation procedures used; this highlights the importance of context-aware evaluation.
- Notably, demographic metadata reveals consistent disparities in performance across all models; this finding emphasizes the need to address systemic biases to ensure equitable outcomes for users with diverse backgrounds.
- While within a model family, accuracy scales with an increase in size, it is not a reliable predictor of downstream accuracy when considering the entire set of models—a crucial observation for future model developers and researchers.

**Audience:**

Yes

**Broader Impact Concerns:**

I am satisfied with the paper, especially with the "What is missing" section and the discussion offered there.

**Claims And Evidence:**

Yes

**Requested Changes:**

- Section 4.6: Please share details on what are the prompts used here. How did you identify said prompts and how did you identify how many times you needed to prompt the model to compute bias metrics?
- Section 4.9.1: I am not sure you can say "should be of right order of magnitude". The difference between power used by data centers powered by nuclear or hydro energy versus those by fossils will be quite dramatic---and that is something you will also see between models trained outside the US (say in France where they primarily use nuclear power) and the US.
- Section 5.5 (Datasets): Please add more detail on who did this manual clustering, what was the procedure, and what was the inter rater agreement.
- Page 46: I don't see orange dots (referred to in the caption) anywhere in the figure.
- Page 50, last paragraph: My read of Figure 29 is that it seems to be a decent predictor, did you mean Figure 30? If not, can you elaborate?
- Page 71: I would like to see the inter rater agreements here. Additionally, please share what quality control mechanisms were in place and what onboarding was provided to the crowd workers.

Typos:
- Page 35 (State Tracking): "performance on (bAbI; Weston et al., 2015)" ---> "performance on bAbI (Weston et al., 2015)"
- Page 60 (Second paragraph): "We find that text-davinci-002 only minor" ---> "We find that text-davinci-002 _presents/experiences_ only minor"
- Page 60 (Summarization): "ROUGE scores tend to much lower" ---> "ROUGE scores tend to _be_ much lower"
- Page 60 (Summarization): "we did find the ROUGE-2 scores did correlate" ---> "we did find _that_ the ROUGE-2 scores ~did~ correlate"
- Page 60 (Summarization): "Since the summarization scenarios were the scenarios that required the longest-form" ---> "Since the summarization scenarios ~were the scenarios that~ required the longest-form"
- Page 61 (Third paragraph): "Notably,the" ---> "Notably, the"
- Page 63: "RAFT helps speaks" ---> "RAFT helps speak"
- Page 72 (Second paragraph): "have to be diverse - if there is" ---> "have to be diverse---if there is" (use an em dash instead of a hyphen)
- Up until page 38, all the quotation marks appear correctly but then there's a shift starting page 39. Please correct them. The Appendix too has mixed quotations everywhere.

**Strengths And Weaknesses:**

### Strengths:
- The paper is very well motivated, really well written, and well situated in literature. It appears the authors have taken a lot of time to get to this stage and that is reflective in the paper.
- The paper presents a thorough and organized effort towards designing HELM. The tradeoffs that have emerged as a result provide very useful insights and I believe this benchmark could be very beneficial for other researchers. The development of a taxonomy for language model evaluation scenarios and metrics makes this work systematically focused and allows for explicit benchmark design limitations, which would allow other researchers to systematically build on top of this benchmark to advance LLM evaluation even as newer LLMs may obtain near perfect numbers on HELM scenarios.
- The implementation of 16 core scenarios along with 7 targeted evaluations provides an extensive benchmarking scope, offering a greater understanding of LMs' capabilities and limitations. Additionally, evaluating 30 models under unified conditions enhances comparability across a range of scenarios and metrics, leading to valuable insights about language models trends.

### Weaknesses:
- The biggest weakness here is that the authors solely focus on English language models. This limitation substantially narrows the scope of evaluation and omits important considerations related to multilingual and diverse dialect models. Expanding beyond the English linguistic domain can contribute to better fulfilling their goal of holistic evaluation.
- The paper relies primarily on automated metrics for evaluation; considering human raters' judgments would provide invaluable insights into the subjective aspects of language model behaviors. Human evaluation, particularly for scenarios where human judgements or fairness and societal impact are of primary concern, could highlight potential pitfalls that might not be captured by automated metrics.
- These large language models exhibit varying performance depending on different prompting decisions and adaptation strategies, leading to potential challenges in standardizing LM evaluation fairly across models. However, sophisticated strategies like prompt-tuning or prompt decomposition are not explored in-depth in this paper, which may inadvertently affect certain findings.
- The benchmark covers "what", "when", and "who" scenarios but appears to miss out on the "why" scenarios, which are equally as important if not more.
- There are several places in the paper where additional details are needed as I share below.

---

> ### Author Response · Authors · 2023-04-27
> **Response to Review gN7D**
>
> Thanks for the thorough and precise review, we appreciate it, especially given the length of this paper! We address each of the weaknesses and requested changes below.
>
> *Weaknesses*
> > The biggest weakness here is that the authors solely focus on English language models ...
>
> We agree that our evaluation is English-dominated, continuing an unfortunate trend in NLP of centering English, which we explicitly name in our work. With that said, this limitation is already being addressed as HELM evolves, beginning with various MT datasets (e.g. WMT, FLORES) and multilingual evaluations.
>
> > The paper relies primarily on automated metrics for evaluation; considering human raters' judgments would provide invaluable insights into the subjective aspects of language model behaviors. ...
>
> We do provide human evaluations for summarization and disinformation, but highlight these are quite costly given the number of models, so we limit these evaluations to cases where we believe there is no sensible automated proxy. Future works have extended HELM with extensive human evaluations; see [Evaluating Human-Language Model Interaction](https://arxiv.org/abs/2212.09746).
>
> > These large language models exhibit varying performance depending on different prompting decisions and adaptation strategies, leading to potential challenges in standardizing LM evaluation fairly across models. However, sophisticated strategies ... are not explored in-depth ...
>
> We agree, however we note (i) we explicitly foreground this limitation is several places, (ii) we simply lack the resources to extensively address this at this time, and (iii) we do provide ablations of various aspects of prompt design in 8.2. More broadly, we might expect that more capable models in the future will be more robust to these prompting details (e.g. GPT-4 often does not require chain-of-thoughts explicitly to perform reasoning tasks effectively).
>
> > The benchmark covers "what", "when", and "who" scenarios but appears to miss out on the "why" scenarios, which are equally as important if not more.
>
> We do not quite understand what "why" refers to, so clarification would be helpful. As we understood, we could think about the reason for why we are interested in a specific scenario (e.g. scientific interest, commercial value). To an extent, we already implicitly include this in our factorization into "core" scenario vs. "targeted" scenarios.
>
> *Requested Changes.*
> > Section 4.6: Please share details on what are the prompts used here. How did you identify said prompts and how did you identify how many times you needed to prompt the model to compute bias metrics?
>
> In 4.6 on fairness, we do not quite follow what is desired, but are happy to provide more details. The code and perturbed examples are available on the public interface, but there are no "prompts" specifically written to elicit fairness or bias. To clarify, for fairness, these are automatic perturbations applied to existing test instances. For bias, there are no changes applied; the evaluation is strictly to the model generations.
>
> > Section 4.9.1: I am not sure you can say "should be of right order of magnitude". The difference between power used by data centers powered by nuclear or hydro energy versus those by fossils will be quite dramatic---and that is something you will also see between models trained outside the US (say in France where they primarily use nuclear power) and the US.
>
> Thanks for raising this point, though we believe there is a misunderstanding here. We agree the factors may differ significantly across regions, but we are referring to the _estimation_ error (as the remainder of the sentence you quote states), not _misspecification_.
>
> That is, the regional carbon intensity of France (the available value) may be off from the specific carbon intensity of the Jean-Zay cluster (used to train BLOOM), but we expect this deviation to be small within regions (and, in general, we are unaware of any finer-grained public data). *This is what we mean when we say "right order of magnitude".*
>
> If we do not know the location of the cluster, we simply do not provide an estimate of emissions due to lack of information.
>
> > Section 5.5 (Datasets): Please add more detail on who did this manual clustering, what was the procedure, and what was the inter rater agreement
>
> The procedure was conducted by the the section's lead author (who is attributed in the public version by name for this section, but we don't mention for anonymity reasons), documented in the corresponding section (E.5) as mentioned in the main paper, and we provide no estimate for inter-rater agreement as a result since the clusters and number of clusters were adaptively constructed.
>
> > Additionally, please share what quality control mechanisms were in place and what onboarding was provided to the crowd worker
>
> The quality control procedure was described on page 71; the annotation guidelines linked in Appendix D.5
>
> > Various typos
>
> All are addressed, thanks!

---

> > ### Author Response · Authors · 2023-04-28
> > **Inter-rate agreement for disinformation annotation**
> >
> > To the requested change on what the inter-rate agreement is, we compute Krippendorff $\alpha$ values for each annotation item.
> >
> > **Reiteration.**
> > 1. support thesis: 0.593
> > 2. style: 0.576
> >
> > **Wedging.**
> > 1. address audience: 0.536
> > 2. support goal: 0.489
> > 3. style: 0.399
> > 4. divisive: 0.257
> > 5. toxic: 0.510

---

> > > ### Comment · Reviewer_gN7d · 2023-05-21
> > > **Appreciate the response**
> > >
> > > - We do not quite understand what "why" refers to, so clarification would be helpful. As we understood, we could think about the reason for why we are interested in a specific scenario (e.g. scientific interest, commercial value). To an extent, we already implicitly include this in our factorization into "core" scenario vs. "targeted" scenarios.
> > >
> > > I was thinking more about core reasoning/causal discovery use cases though that's not a major issue here.
> > >
> > > I would also appreciate clarification in Section 5.5 that it was one of the authors who did these annotations (and the limitations that are associated with such an annotation framework).

---

> > ### Public Comment · ~JyothiSwaroopReddy_Bommareddy1 · 2023-08-26
> > **nice good**
> >
> > this review was useful 0iub kk i i

---

### Author Response · Authors · 2023-04-28
**Author response to reviews**

We thank the reviewers for their extensive, acute, and incisive feedback on the paper. We understand this paper is especially long, so we are grateful for all of their effort, in addition to the Action Editor for facilitating the review process.

We have provided direct responses or made updates to address all weaknesses or change requests from the reviewers. All three reviewers highlight a number of shared strengths for the work, so we take this as an encouraging sign.

At the top level, when addressing change requests/weaknesses, we highlight three common themes.

1. *The paper is difficult to navigate.* To address this, we highlight we will add a table-of-contents (subject to approval from EiCs) and the public interface also significantly improves the accessibility of the results.

2. *The evaluation could be more extensive in some dimension: languages beyond English, adaptation beyond prompting, evaluation beyond automatic metrics*. These are all excellent and well-taken points that we foreground in our writing as limitations. With that said, while largely these were not addressed as a matter of cost, we note several of them already have been addressed in the live version of the HELM benchmark or in other works that built on HELM. Overall, since all reviewers cite the work as quite comprehensive, we hope this can be fairly seen as out-of-scope for this work and precisely the type of future work that HELM can support.

3. *The evaluations, while well-designed, do not track many of the new use cases for LMs (especially post ChatGPT/GPT-4)*. We note the work was conducted in 2021/early 2022 and released prior to both ChatGPT and GPT-4, when the actual deployment of LMs was not as prevalent as present. With that said, much as in 2, we believe this can be addressed in the future versions of HELM (in the sense that ongoing efforts aim to address this).

Overall, we highlight that HELM is a unique work to evaluate, since the paper in question is about a static snapshot of HELM but some of the limitations in 2-3 can/will be addressed by its status as a living artifact. Consequently, we acknowledge these are fair criticisms of HELM the paper, but note their severity can be blunted because there is a clear sense in which they can all be addressed (with some of the subsequent versions of HELM already addressing them).

---

### Decision · Action_Editors · 2023-06-26

**Recommendation:** Accept as is

**Comment:**

This paper presents a truly holistic evaluation for large language models (a "first of its kind study in terms of scope" as one reviewer put it), proposing several axes along along which the research community can benchmark them. The authors perform extensive experiments with state-of-the-art LMs (both open-source and closed ones) and present a large array of core findings ranging from issues like memorization, toxicity, to the relationships between robustness, fairness and accuracy. As noted by the reviewers, the claims made by the paper are convincingly supported in the experiments. The authors also revised the paper following reviewer suggestions in order to make the paper easier to navigate. I'm confident this will paper will serve as a key reference for future research into the design and evaluation of LLMs in terms of both concrete findings and the general methodologies proposed. Given its comprehensiveness and usefulness to the community, I also recommend a Featured Certification.

In your camera ready version, please add a table of contents to help readers. This should be included in the main body, after the introduction.

**Audience:**

Yes, this paper (and associated benchmark that is continually being updated) will substantially influence future work in large language model, and the NLP and ML communities at large.

**Claims And Evidence:**

The paper presents substantial empirical evidence to support the proposal of providing a holistic evaluation of large language models. The reviewers appreciated the large-scale effort and felt the writing was clear, well motivated and comprehensive.